



# A new method for long-term source apportionment with time-dependent factor profiles and uncertainty assessment using SoFi Pro: application to one year of organic aerosol data

Francesco Canonaco[1,2], Anna Tobler[2], Gang Chen[2], Yulyia Sosedova[1], Jay Gates Slowik[2], Carlo Bozzetti[1,2], Kaspar Rudolf Daellenbach[3], Imad ElHaddad[2], Monica Crippa[4], Ru-Jin Huang[5], Markus Furger[2], Urs Baltensperger[2], André Stephan Henry Prévôt[2]

[1]Datalystica Ltd., Park innovAARE, CH-5234 Villigen, Switzerland
[2]Paul Scherrer Institute, Laboratory of Atmospheric Chemistry, CH-5232 Villigen PSI, Switzerland
[3]Institute for Atmospheric and Earth System Research, Helsinki, Finland
[4]European Commission, Joint Research Centre (JRC), Via Fermi, 2749, 21027 Ispra, Italy
[5]State Key Laboratory of Loess and Quaternary Geology, Center for Excellence in Quaternary Science and Global Change, and Key Laboratory of Aerosol Chemistry and Physics, Institute of Earth Environment, Chinese Academy of Sciences, Xi'an 710061, China

*Correspondence to*: Canonaco F., (francesco.canonaco@psi.ch)

**Abstract.** A new methodology for performing long-term source apportionment (SA) using positive matrix factorization (PMF) is presented. The method is implemented within the SoFi Pro software package and uses the multilinear engine (ME-2) as a PMF solver. The technique is applied to a one-year aerosol chemical speciation monitor (ACSM) dataset from downtown Zurich, Switzerland.

The measured organic aerosol mass spectra were analyzed by PMF using a small (14 days) and rolling PMF window to account for the temporal evolution of the sources. The rotational ambiguity is explored and the uncertainty of the PMF solutions were estimated. Factor/tracer correlations for averaged seasonal results from the rolling window analysis are higher than those retrieved from conventional PMF analyses of individual seasons, highlighting the improved performance of the rolling window algorithm for long-term data.

In this study four to five-factors were tested for every PMF window. Factor profiles for primary organic aerosol from traffic (HOA), cooking (COA) and biomass burning (BBOA) were constrained. Secondary organic aerosol was represented by either the combination of semi-volatile and low-volatility organic aerosol (SV-OOA and LV-OOA, respectively), or by a single OOA when this separation was not robust. This scheme leads to roughly 40'000 PMF runs. Full visual inspection of all these PMF runs is unrealistic and is replaced by predefined user-selected criteria, which allow factor sorting and PMF run acceptance/rejection. The selected criteria for traffic (HOA) and biomass burning (BBOA) were the correlation with equivalent black carbon (eBC$_{tr}$) and the explained variation of $m/z$ 60, respectively. COA was assessed by the prominence of a lunchtime concentration peak within the diurnal cycle. SV-OOA and LV-OOA were evaluated based on the fraction of $m/z$ 43 and $m/z$ 44 in their respective factor profiles. Seasonal *pre*-tests revealed a non-continuous separation of OOA into SV-OOA and LV-



OOA, in particular during the warm seasons. Therefore, a differentiation between four-factor solutions (HOA, COA, BBOA
and OOA) and five-factor solutions (HOA, COA, BBOA, SV-OOA and LV-OOA) was also conducted based on the criterion
for SV-OOA.

HOA and COA contribute between 0.4-0.7 µg·m⁻³ (7.8-9.0 %) and 0.7-1.2 µg·m⁻³ (12.2-15.7 %) on average throughout the
year, respectively. BBOA shows a strong yearly cycle with the lowest mean concentrations in summer (0.6 µg·m⁻³, 12.0 %),
slightly higher mean concentrations during spring and fall (1.0 and 1.5 µg·m⁻³, or 15.6 and 18.6 %, respectively), and highest
mean concentrations during winter (1.9 µg·m⁻³, 25.0 %). In summer, OOA is separated into SV-OOA and LV-OOA, with
mean concentrations of 1.4 µg·m⁻³ (26.5 %) and 2.2 µg·m⁻³ (40.3 %), respectively. For the remaining seasons the seasonal
concentrations of SV-OOA, LV-OOA and OOA range from 0.3-1.1 µg·m⁻³ (3.4-15.9 %), 0.6-2.2 µg·m⁻³ (7.7-33.7 %) and 0.9-
3.1 µg·m⁻³ (13.7-39.9 %), respectively. The relative PMF errors modelled for this study for HOA, COA, BBOA, LV-OOA,
SV-OOA and OOA are on average ± 34 %, ± 27 %, ±30, ±11 %, ±25 % and ±12 %, respectively.

## 1 Introduction

Atmospheric aerosols are at the center of scientific and political air quality discussions due to their highly uncertain direct and
indirect climate effects (IPCC, 2013) and negative impact on human health (e.g., Peng et al. (2005)). Regulatory policies
addressing these effects require characterization and understanding of aerosol physicochemical properties, sources and
formation processes. During the past years, the study of submicron particulate matter (PM$_1$) has gained interest (Hallquist et
al., 2009), in particular the organic fraction comprising 20-90% of the total submicron aerosol mass (Jimenez et al., 2009).
Atmospheric aerosols are typically classified as primary or secondary aerosols, where primary aerosols are directly emitted
into the atmosphere and secondary aerosols are formed by reaction of precursor gases. Aerodyne aerosol mass spectrometers
(AMS) and aerosol chemical speciation monitors (ACSM) have become important and widely used instruments for the on-line
chemical characterization of non-refractory submicron aerosol (NR-PM$_1$) (Canagaratna et al., 2007; Ng et al., 2011b; Fröhlich
et al., 2013). The resulting aerosol data can be utilized to study seasonal trends of PM$_1$ sources to support emission reduction
strategies. This is highly relevant for very polluted areas like China and India but also for Europe, where particulate matter
concentrations substantially decreased during the last two decades, but still frequently exceed legal thresholds (Barmpadimos
et al., 2011; Barmpadimos et al., 2012; European Environment Agency, 2019).

Source apportionment of organic aerosol (OA) measured with an AMS and / or ACSM is typically performed using the positive
matrix factorization algorithm (PMF, Paatero and Tapper (1994)). PMF solutions describe the complex, time-dependent
organic aerosol composition as a linear combination of static factor profiles (for AMS/ACSM data, mass spectra) and their
time-dependent contributions. Factors can represent a primary organic aerosol emission (POA) or secondary organic aerosol
(SOA).



Many organic source apportionment studies with AMS (see review by Zhang et al. (2011)) and ACSM data (e.g., (Aurela et
al., 2015; Budisulistiorini et al., 2013; Canonaco et al., 2013; Fröhlich et al., 2015; Li et al., 2017; Minguillon et al., 2015;
Reyes-Villegas et al., 2016; Ripoll et al., 2015; Schlag et al., 2016; Sun et al., 2013; Sun et al., 2018; Tiitta et al., 2014; Wang
et al., 2017; Zhang et al., 2019; Zhu et al., 2018)) have successfully employed the PMF algorithm. PMF results suffer from
rotational ambiguity (Paatero et al., 2002), i.e., several PMF results exist with a similar goodness of fit. An approximate method
for the quantification of the rotational uncertainty, i.e., the amount of rotational ambiguity (Paatero et al., 2014), can be
obtained using the global *f*peak tool, which allows exploration of a single one-dimensional transect through the
multidimensional solution space and is discussed for AMS data in Ulbrich et al. (2009). This approach only leads to a rough
estimate of the rotational uncertainty, as it allows investigation of only a single transect whose selection is uncontrollable,
while other rotations remain entirely inaccessible. An improved method for both uncertainty estimation and factor resolution
was demonstrated by Canonaco et al. (2013), where intelligent exploration of rotations was implemented introducing *a priori*
information in form of factor profiles in the multilinear engine (ME-2, Paatero (1999)). Moreover, Ulbrich et al. (2009) also
estimated the statistical uncertainty via the resampling bootstrap technique (Efron, 1979). This method generates a set of new
input matrices for analysis from random resampling of the original input data. This resampling perturbs the input data by
including replicates of some points while excluding others, with the main assumption that the overall properties of the analyzed
data (fingerprints of the factors, contributions of the factors) are not systematically changed, i.e., changes are purely statistical.
If a sufficient number of resamples has been carried out, the variation within the identified factors across all bootstrap runs is
regarded to represent their statistical uncertainty.

A crucial limitation of the traditional PMF approach is that the time-dependent variability of the composition of the organic
aerosol sources cannot be properly modelled using static profiles in a year-long PMF model. Both POA and SOA may have
time-dependent composition. For example, vehicles utilize different fuel blends in winter and summer for traffic (Agrola,
2017), while biomass burning may be dominated by different burning types and / or materials in different seasons e.g., domestic
heating in winter, agricultural burning in spring/autumn, wildfires in summer. SOA sources may likewise be affected by
seasonal changes in either precursor emissions (e.g., monoterpene emissions increase exponentially with temperature) or
physicochemical processes (e.g., gas/particle partitioning, oxidant concentrations) (Hallquist et al., 2009). Amongst others,
Canonaco et al. (2015), Daellenbach et al. (2017) and Sun et al. (2018) showed that ACSM SOA mass spectra possess distinct
seasonal trends which need to be considered during the PMF analysis. For Zurich, Stefenelli et al. (2019) and Qi et al. (2019)
were able to demonstrate this seasonal variability of SOA characteristics by molecular analysis, with terpene related SOA
being dominant in summer and aged wood burning organic aerosol being dominant in winter.

Technically, modeling seasonally-dependent mass spectra from a given source family, e.g., traffic, biomass burning, or SOA,
can be achieved in two ways. PMF can be applied to a multi-season data set, with time-dependent source composition
modelling of a single factor per source or source class, similar to typical representations of SOA in short-term field campaigns
by two factors with different degrees of oxygenation (Zhang et al., 2011). However, multi-factor representations of seasonal
changes are likely to significantly increase the complexity of the PMF solution, primarily due to a rapid increase in the number



of factors and thus leading to difficulties in interpretation. Another possibility is to perform PMF over a small, moving time frame such that the factor profiles evolve with time, while maintaining a single factor per source family. This is likely the best

choice for long-term data, due to both the relative simplicity of the solution and important savings in computational and evaluation time. The latter is also more compatible with a continuously growing dataset, e.g. for online source apportionment studies, where the entire dataset doesn't have to be completely reanalyzed when new data is included in contrast to classical batch analyses. Parworth et al. (2015) have already shown the effectiveness of such an approach, i.e., employing a small and moving PMF window for analyzing remote long-term ACSM data with only a few unconstrained aerosol sources / components.

However, a rotational and statistical uncertainty exploration was not conducted.

This study presents the analysis of ACSM data measured in Zurich between February 2011 and February 2012. The dataset includes several sources that were difficult to separate using unconstrained PMF, which are constrained using known POA sources in ME-2 for a small and rolling time window. This strategy allows to adequately account for time-dependent variation of the POA and SOA factor profiles. The applied constraining technique allows for a more comprehensive and quantitative

assessment of the rotational uncertainty than the global $f$peak tool could achieve. The statistical uncertainties of PMF solutions are estimated using a bootstrap resampling technique. In this study, the size of the rolling window, tightness of constraints, and several other parameters as e.g. number of PMF repeats per rolling window, are discussed and validated.

## 2 Instruments and methods

### 2.1 Instrumentation and sampling site

An ACSM (Aerodyne Research, Inc., Billerica, MA, USA) was deployed at the Kaserne station, an urban background station in the city center of Zurich (Switzerland) between February 2011 and February 2012 (Lanz et al., 2007; Lanz et al., 2008; Canonaco et al., 2013). The ACSM is an instrument based on Aerodyne aerosol mass spectrometer (AMS) technology, but optimized for long-term measurements with minimal maintenance requirements. The ACSM measures the real-time

composition of non-refractory submicron particulate matter, customarily referred to as NR-PM$_1$. The instrument is described in detail in Ng et al. (2011b), (see also Jayne et al. (2000), Jimenez et al. (2003), Allan et al. (2003), Allan et al. (2004), and Canagaratna et al. (2007) for a more detailed description of the AMS technique). Technical problems on the ACSM inlet system during the last third of the campaign resulted in a total of 2-3 weeks of missing data.

The ACSM in Zurich was operated with a scan rate of 1 s/amu between $m/z$ 10 and 140, and produced averaged scans every

15 min. The data was re-averaged to 30 min for ME-2 analysis. To obtain quantitative mass concentrations for ACSM data, a collection efficiency parameter (CE) needs to be applied to account for the incomplete detection of aerosol species due to particle bounce at the instrument vaporizer (Middlebrook et al., 2012). The effects of the nitrate mass fraction and particle acidity on CE have been parameterized for ambient data (Middlebrook et al., 2012). As discussed previously (Canonaco et al., 2013; Canonaco et al., 2015) CE = 1 for the current study is assumed because of otherwise systematic overestimation compared





to the PM$_{10}$ measurements by a tapered oscillating microbalance (TEOM, FDMS 8500, Thermo Scientific) calibrated by gravimetric measurements of off-line PM$_{10}$ filters.

The meteorological data (temperature, relative humidity, solar radiation, precipitation, wind speed and wind direction) and trace gases (CO, NO$_x$, O$_3$, total hydrocarbons) were measured by the Swiss National Air Pollution Monitoring Network, NABEL (Empa, 2010). Equivalent black carbon (eBC) was measured with an Aethalometer AE 31 (Magee Scientific Inc.,

Berkeley, CA, USA). The data were corrected for loading effects and multiple scattering using the method of Weingartner et al. (2003). Mass absorption cross sections as determined by Herich et al. (2011) were used to convert $b_{abs}$($\lambda$=880nm) to eBC. The measured absorption coefficients at wavelengths 470 and 880 nm using the alpha-values based on Zotter et al. (2017) were used to estimate the contributions to eBC from traffic (eBC$_{tr}$) and biomass burning (eBC$_{wb}$).

Seasonal PMF runs performed on the ACSM data in earlier studies (Canonaco et al., 2013; Canonaco et al., 2015) showed

three primary OA factors and one to two secondary OA factors contributing throughout the measurement year. Among the primary OA factors a traffic-related hydrocarbon-like organic aerosol (HOA) factor was found, which correlated with NO$_x$ and eBC$_{tr}$, as well as a biomass burning organic aerosol (BBOA) factor, which correlated with eBC$_{wb}$ as also shown in other studies (Lanz et al., 2007; Lanz et al., 2008; Ulbrich et al., 2009; Zhang et al., 2011; Canonaco et al., 2013). Given that in summer the daily values of *m/z* 60 were always higher than the threshold for biomass burning influence identified in Cubison

et al. (2011), BBOA was also modelled during the warm seasons. The third primary OA factor was assigned to cooking-related organic aerosol (COA), and exhibited enhanced concentrations during mealtimes, similar to previous studies (Allan et al., 2010; He et al., 2010; Slowik et al., 2010; Sun et al., 2011; Mohr et al., 2012; Crippa et al., 2013; Elser et al., 2016). For warm days during the first winter and in spring, summer and fall the variability of the bulk OOA (oxygenated organic aerosol) was captured by two distinct factors, i.e., SV-OOA (semi-volatile oxygenated organic aerosol) and LV-OOA (low-volatility

oxygenated organic aerosol). For the remaining colder period only one OOA factor accounted for the variation of the bulk OOA.

### 2.2 Methods

#### 2.2.1 The multilinear engine (ME-2)

ME-2 (Paatero, 1999) is a powerful engine for solving the positive matrix factorization algorithm (PMF, (Paatero and Tapper,

1994)). Model configuration and post-analysis are performed by Source Finder (SoFi Pro 6.8, Datalystica Ltd., Villigen, Switzerland) within Igor Pro software environment (Wavemetrics, Inc., Portland, OR, USA) as described in Canonaco et al. (2013). In its bilinear mode, PMF describes the measured data matrix **X** as a product of two matrices, **G** and **F** and the residual matrix **E**. In element notation the equation is:

$$x_{ij} = \sum_{k=1}^{p} g_{ik} \cdot f_{kj} + e_{ij} \tag{1}$$

In the measured matrix **X** the columns *j* are the *m/z*'s and each row *i* represents a single mass spectrum. *p* is defined as the number of factors of the selected model solution, i.e., the number of columns of **G** and the number of rows of **F**. Each column





of the matrix $\mathbf{G}$ represents the time series of a factor, whereas each row of $\mathbf{F}$ represents the factor profile (i.e., mass spectrum); both are indexed by $k$. In an unconstrained PMF run in ME-2, the model is initialized with random entries in $\mathbf{G}$ and $\mathbf{F}$ ("seed") and the quantity $Q$ is minimized with respect to all model variables by means of the conjugate gradient algorithm (Paatero,

165   1999):

$$Q = \sum_{i=1}^{n} \sum_{j=1}^{m} \left(\frac{e_{ij}}{\sigma_{ij}}\right)^2 \tag{2}$$

where $e_{ij}$ are the elements of the residual matrix $\mathbf{E}$ and $\sigma_{ij}$ represents the measurement uncertainty for the input point $x_{ij}$.

To compare $Q$-values from various PMF runs with a different size and / or number of factors, $Q$ is normally scaled by the remaining degrees of freedom ($Q_{exp}$, which depends on the size of the input data and the number of chosen factors):

$$Q_{exp} = m \cdot n - p(m + n) \tag{3}$$

PMF is subject to rotational ambiguity, in which different combinations of $\mathbf{G}$ and $\mathbf{F}$ yield similar $Q$-values. Some of these combinations may contain mixed factors and / or environmentally unreasonable descriptions of the data. Previous work has shown that constraining expected factor profiles using the $a$-value approach for AMS/ACSM data is an efficient method for isolating the set of environmentally interpretable PMF runs (Lanz et al., 2008; Canonaco et al., 2013; Crippa et al., 2014). The

$a$-value determines the extent to which the $m/z$ in the mass spectral profile, also referred to as anchor ($f_{kj}$), is allowed to vary during the model iteration according to:

$$f_{kj}{'} = f_{kj} \pm a \cdot f_{kj} \tag{4}$$

The index $j$ stands for the actual variable ($m/z$) of the $k^{\text{th}}$ factor, and the $a$-value is its scalar product. For example, an $a$-value of 0.1 allows for a variability of $\pm$ 10% during the iterative process. This implies that some variables might increase and some

might decrease within this limit. Note that after renormalizing the solution, the extent to which the constrained values changed might be slightly outside the defined $a$-value range. For example, consider a case where the $a$-value is set to 0.1 for all variables of a factor profile. The values of all variables but one could decrease by 10% while the value of this single variable might increase by 10% during the iteration. After renormalizing the factor profile outside ME-2 by, e.g., the sum of the profile, the intensity of this single variable will exceed the boundaries set with the $a$-values during the PMF iteration. Moreover, note that

the $a$-value approach defines only the boundaries of a solution and does not imply any weighting within these boundaries.

### 2.2.2 PMF input preparation step

The organic data and error matrices (Allan et al., 2003) are computed using the ACSM local tool version 1.5.3.2 (Aerodyne Research, Inc., Billerica, MA, USA) in Igor Pro. Weak (signal to noise ratio between 2 and 0.2) and bad variables (signal to noise below 0.2) were downweighted according to the recommendations in Paatero and Hopke (2003). The $m/z$ 16, 17, 18 and

28 variables that are replicates of the variability of $m/z$ 44 were removed for the PMF calculation and recalculated *a posteriori* as a function of the $m/z$ 44 contribution attributed to each factor profile (Elser et al., 2016). This approach is preferable to



downweighting (Ulbrich et al., 2009), as it maintains a direct mathematical relationship between m/z 44 and its dependent variables, which can otherwise be distorted by dynamic weighting of outliers in the PMF robust mode.

### 2.3 New rolling method using ME-2

The new method consists in performing PMF runs on a small and moving window that is translated across the entire dataset. At each step, many individual PMF runs are performed, and the resulting runs are accepted or rejected according to predefined criteria. The window is then moved to the next position, with the distance between window positions being significantly smaller than the window size itself. The set of all accepted PMF runs determines the final source apportionment solution and is also used to assess model uncertainties.

The novelty of this method compared to Parworth et al. (2015) lies in the application of ME-2 for enhanced control over the matrix rotations, and in the automated application of user-defined criteria to determine the set of accepted PMF runs. Moving properties of the window (window-runs) are discussed in Section 0, whereas the main settings of PMF within a window (PMF runs) are described in Section 0.

### 2.3.1 The rolling strategy

PMF analysis is conducted on a subset of data defined by a small window that is moved in 1-day increments across the entire dataset and as such allows capturing seasonal variations of the factor profiles. Note that rolling windows containing less than 10 % of real-data are automatically skipped by the rolling algorithm. This avoids performing PMF runs over large gaps due to, e.g., calibrations or instrument failures. The window size ($s_{win}$) is a free parameter that requires optimization. The rolling window PMF analysis of Parworth et al. (2015) utilized a 2-week window, arguing that this length is representative of the

average lifecycle of aerosols in the atmosphere. Even for (low time-resolution) ACSM data, two weeks have been shown to provide enough temporal variability to distinguish sources with similar factor profiles such as HOA and COA (Fröhlich et al., 2015) In the present study, likewise a 14-day window is selected, after additionally assessing the performance of 3, 7, 21, and 28-day windows.

The model performance in response to $s_{win}$ is assessed by monitoring the value of $Q/Q_{exp}$ (which decreases as the mathematical

goodness of fit improves) and the number of non-modelled time-points ($n_{\text{non-modelled}}$) as a percentage of the total number of measurements. $n_{\text{non-modelled}}$ is defined as any ACSM time-point for which the user-defined criteria (see Sections 0 and 0) are not met for any PMF runs that include this measurement (note that for most points this will include PMF runs from multiple overlapping windows). Figure 1a shows $Q/Q_{exp}$ and $n_{\text{non-modelled}}$ as a function of $s_{win}$. The $Q/Q_{exp}$ values are minimized for a 7-day window and are approximately 15 % higher for the 3- and 14-day windows, and 45 % higher for the 21- and 28-day

windows. $n_{\text{non-modelled}}$ shows a minimum for 14 days with a slight increase for larger windows and a steep increase for smaller $s_{win}$.

A 14-day window has been chosen for the current dataset, as this avoids significant increases in $Q/Q_{exp}$ without inducing unacceptably high $n_{\text{non-modelled}}$. Moreover, because the 1-day step of the rolling window is smaller than the 14-day width, each





time-point is included in 14 different window-runs (except for those within the first or last 14 days of the dataset). As discussed later, these repeats aid the uncertainty analysis.

### 2.3.2 Window settings

The rolling strategy described above defines a new window after every window shift. Within this new window, a PMF run, referred to as repeat in the text, is generated via ME-2, which initializes new seeds, $a$-values, and bootstrap resampling of the PMF input. The seed initializes all model entries in **G** and **F**, and unconstrained information therein is randomly initialized.

Additionally, *a priori* information on the factors from the seasonal *pre*-tests is used to confine the solution space and thus to decrease the rotational ambiguity of the solution.

In the current study, constraints are applied only to profiles of the POA factors, namely traffic (HOA), cooking (COA) and biomass burning (BBOA). The HOA and COA profiles are taken from Crippa et al. (2013), while BBOA is the averaged mass spectrum reported by Ng et al. (2011a). These anchor profiles were also successfully used for the seasonal analysis of the

Zurich-Kaserne data (Canonaco et al., 2013; Canonaco et al., 2015).

Every constrained factor profile applied in a PMF run requires a sensitivity analysis of the $a$-value to identify the range of reasonable solutions (Canonaco et al., 2013; Crippa et al., 2014; Elser et al., 2016). Typically, variation of the $a$-value of one or more constrained factor profile(s) allows exploration of a region in the solution space that includes environmentally reasonable solutions. In the present analysis, the goal is to consider all PMF runs (not just the best one) that are mathematically

and environmentally reasonable. Recent studies have systematically investigated the entire solution space allowed by the $a$-values, e.g. by conducting PMF runs covering every combination of $a$-values over the range 0 to 1 with a step of 0.1 (Elser et al., 2016; Bozzetti et al., 2017; Daellenbach et al., 2017). However, this approach is not computationally practical for moving window analysis. For instance, given that three factors are constrained in this present study, the above $a$-value exploration strategy would require $11^3 = 1331$ PMF runs for $a$-value exploration per window-run. Also, each combination of $a$-values

would require a minimum of 100 PMF runs for bootstrap analysis (Norris et al., 2014). Furthermore, the seasonal *pre*-tests indicated that both four- and five-factor solutions should be assessed (corresponding to one or two OOA factors). In total, this would require $1331 \times 100 \times 2 \sim 2.66 \times 10^5$ PMF runs per window. Moreover, the daily shift of the rolling window will initialize the window-runs 351 times (one year minus the $s_{win}$), resulting in $1331 \times 100 \times 2 \times 351 \sim 9.35 \times 10^7$ PMF runs for a systematic analysis. This will require several months of computation even on modern PCs with multi-core processors. To overcome these

computational issues, two strategies were considered for reducing the number of runs required for $a$-value exploration. In both cases, a systematic exploration of the $a$-value space is replaced by randomly generated $a$-values between zero and an upper limit ($a_{max}$). For the first strategy, the $a_{max}$ limit was fixed at one, and the number of repeats ($x_{PMF}$) was adjusted until the same criteria described above for $s_{win}$ optimization were satisfactory. However, this approach was rejected, as executing the full set of PMF runs required computational times on the order of months (see supplement A) and therefore was impractical on regular

PCs.

The second strategy, which is used here, exploits the *a priori* information of the sources. If some factor profiles are known to be present and their source profiles are known to some extent, there is no need to explore regions in the solution space, for which these factor profiles may drastically depart from their realistic anchors.

Therefore, $a_{max}$ undergoes a systematic scan from zero upwards, with model performance assessed by $Q/Q_{exp}$ and $n_{non\text{-}modelled}$,
as described above for the $s_{win}$ estimation. The current strategy counts as local-minimum algorithm, as the full parameter space ($s_{win}$, $a_{max}$, $x_{PMF}$) is not fully investigated. Moreover, *pre*-tests based on literature data, i.e. a 14-day PMF window for $s_{win}$ (Parworth et al., 2015) and an upper $a$-value of 0.3 $a_{max}$ (Crippa et al., 2014) represented the starting condition for the parameter optimization discussed in Figure 1.

Figure 1b shows an almost flat $Q/Q_{exp}$ while that of the $n_{non\text{-}modelled}$ behaves as a quadratic function with a minimum at $a = 0.4$.
For $a$-values below 0.4 the constrained fingerprints cannot optimally adapt to the current input. Given only 50 random $a$-value explorations out of 1331 (see above) of the entire $a$-value space for every PMF window, outcomes for higher $a_{max}$ may be purely stochastic and lead to a high degree of mixing and consequently rejection of the PMF runs (high $n_{non\text{-}modelled}$). $a = 0.4$ represents the optimum $a_{max}$ and is set as free parameter for the $a$-value exploration.

The random resampling of the PMF input uses the bootstrap approach for every repeat. A window comprising 14 days with at
most 48 (number of scans per day) x 14 (days)= 672 time-points will create resamples containing again 672 new time-points, where some time-points may occur multiple times and others may be absent. As above, $Q/Q_{exp}$ and the percentage of $n_{non\text{-}modelled}$ are monitored as a function of the $x_{PMF}$. Figure 1c reveals a constant $Q/Q_{exp}$ whereas the number of $n_{non\text{-}modelled}$ decreases and stabilizes from 50 repeats onwards. We conclude that 50 repeats per window are sufficiently high for the bootstrap strategy. Note that the final number of PMF runs per time-point may be higher than $x_{PMF}$ due to the overlapping PMF runs resulting
from the rolling strategy. The total number of PMF runs for this study equals 50 ($x_{PMF}$) x 351 (number of days) x 2 (four- and five-factors) = 35'100 runs and required approximately three days on a modern multicore PC.

### 2.3.3 The *post*-PMF analysis

Manual inspection of all generated PMF runs is impractical, and is replaced by an automated procedure based on pre-defined user criteria that (1) identifies and sorts unconstrained factors and (2) determines whether each PMF run should be accepted
or discarded. Examples of user-defined criteria could include the factor correlation to an external tracer in terms of either the overall time series or diurnal pattern, or characteristic temporal features, e.g., a prominent lunch peak for a cooking factor. Modelled PMF factors for which no factor criteria are satisfied, i.e. very poor score values due to factor mixing / swapping or sampling of a transient sources not accounted for, typically yield $n_{non\text{-}modelled}$.

In addition to determining whether an individual PMF run should be accepted or rejected, the criteria are used to determine
the identity of unconstrained factors. While the positions of constrained factors within the **F** and **G** matrices are *pre*-defined for constrained factors, the same is not true of unconstrained factors, and these must be correctly identified prior to further data analysis. Consequently, all possible combinations for sorting unconstrained factor positions are evaluated (factor identification) and their scores combined together. As criteria with various score ranges are potentially possible, e.g.,



correlation coefficient, lunch peak ratio, the explained variation (EV, see Eq. 5) of *m/z* 60 and variable fractions, these score

values must be corrected before being added up. z-score transformation as a linear correction is applied, where at the end the score distribution of each criterion possesses a mean value of zero and a standard deviation of one. Finally, the z-score transformed combination with the highest values is chosen to represent the PMF result for a specific PMF run. This is essential in the case of the two unconstrained factors SV-OOA and LV-OOA in this study. Note that this requires criteria to be defined for a minimum of all factors but one (i.e., $p$ -1 factors).

Considering the large amount of PMF runs by the rolling window algorithm, the main advantage of this criteria-based inspection is that the complexities of a factor profile and time series are reduced to single values ("scores"). Based on the score plots, potentially promising PMF runs can be further investigated and validated. This significantly improves the efficiency of PMF analysis by discarding PMF runs where the score for any criterion falls below the user-defined threshold ("bad PMF runs"). In contrast to conventional analyses, where a single PMF run often represents an optimal description of the dataset, the

entire set of PMF runs classified as environmentally reasonable is used for the analysis and presentation. This provides a more comprehensive and robust representation of the dataset and supports uncertainty assessment.

    To determine whether an individual PMF run is accepted or rejected, acceptance thresholds are defined for each of the selected criteria. These thresholds are free parameters and must be defined for each criterion separately. Either a threshold is inferred from previous studies or from significance tests or similar statistical analyses (see discussion for the HOA and COA thresholds

in Section 0 for such an example).

    The computational time required for criteria application subsequent averaging is typically on the order of minutes to hours with a modern multicore PC, depending on the amount of accepted PMF runs. Thereafter, the results can be inspected in real-time allowing the user to efficiently investigate the set of PMF runs and if needed, test various criteria.

### 2.3.4 Chosen criteria in this study

In this study one criterion per factor was defined, although it is possible to apply multiple criteria to the same factor, as each criterion is assessed individually on an accept/reject basis.

    Figure 2 shows the criterion scores calculated for each PMF run, with each plot representing an individual factor. The gray points show the score values for all PMF runs, the blue points denote PMF runs where criterion thresholds are satisfied, and the green points represent PMF runs where criterion thresholds for all criteria are simultaneously fulfilled. These green points

are then used to compute the final PMF solution. The criteria and their corresponding thresholds applied for each criterion (blue points in Figure 2) are also reported in Table 2 (1[st] value).

    In the current study, the thresholds for the criteria of HOA and COA were determined based on statistical analyses with the help of the results from conventional (no rolling technique) seasonal PMF from previous studies (Canonaco et al., 2013; Canonaco et al., 2015). The contribution of HOA and its tracer $eBC_{tr}$ were bootstrapped together and the correlation coefficient

($R_{Pearson}$) was evaluated each time, leading to a distribution for $R_{Pearson}$. Similarly, the time series of COA was bootstrapped and the lunch peak enhancement in COA evaluated each time ($COA_{11+12+13hrs}/COA_{9+10+14+15hrs}$), leading to a distribution for the



lunch peak concentration. Finally, the 10th percentile value was chosen as threshold score value. These seasonal thresholds are also visible as steps in the score plots (blue points in Figure 2 a) and b), respectively) and are also reported in Table 2 (2nd value in brackets). For spring 2011, summer 2011 and winter 2012 however, the resulting thresholds for HOA either caused

too many missing time-points ($R_{Pearson} = 0.8$) or had rather non-significant correlation coefficients ($R_{Pearson} = 0.2$, with a $p$ value of 0.4, $n = 24$ as for the other seasons). Hence, these thresholds were systematically lowered for spring 2011 and increased for winter 2012 to achieve the highest possible correlation coefficient with maximal data coverage, i.e. same $n_{non-modelled}$ when considering all PMF runs for these periods in these criteria.

NO$_x$ is a typical tracer for HOA in urban areas. However, due to incomplete NO$_x$ measurement coverage in this campaign

(especially during spring and fall), eBC$_{tr}$ is used as a traffic tracer and the $R_{Pearson}$ correlation coefficient is computed between the diurnal cycle of eBC$_{tr}$ and the HOA factor.

As is frequently the case, no chemical tracers for COA were available in this study. Previous measurements in Zurich (Canonaco et al., 2013; Canonaco et al., 2015) have demonstrated a strong diurnal pattern for COA, with an increased concentration during lunchtime. As a proxy for COA, the lunch-time COA enhancement is monitored (Table 2).

The wood burning contribution to black carbon (eBC$_{wb}$) as determined by the eBC source apportionment (eBC-SA) method of Sandradewi et al. (2008) was considered as a possible criterion for BBOA but then rejected. The eBC-SA analysis applies to air masses highly influenced by biomass burning and has been validated for winter data only. Uncertainties in eBC$_{wb}$ during warm seasons, when the biomass burning contribution is small, have been shown to be quite high (Harrison et al., 2013). Therefore, it was decided to use another metric for BBOA, exploiting the key spectral feature at $m/z$ 60. For BBOA the

explained variation (EV) (Paatero, 2010) for $m/z$ 60 is monitored as follows:

$$EV_{j,k} = \frac{\sum_{i=1}^{n}(|g_{ik} \cdot f_{kj}|/\sigma_{ij})}{\sum_{i=1}^{n}((\sum_{h=1}^{p}|g_{ih} \cdot f_{hj}| + e_{ij})/\sigma_{ij})} \tag{5}$$

This threshold is chosen following the recommendation in Paatero (2010), where a variable modelled by its mean explains already ~25% of the variation. If the measured variability of a variable is explained by a specific factor, that factor must capture more than the mean value of the variable, and hence Paatero (2010) recommended 30-35 % as a minimum EV. However, using

30 or 35 % as threshold resulted in several weeks of non-modeled time-points in particular for spring and fall 2011. $a$-value of 25 % resulted in a reasonable compromise between EV and the amount of non-modeled time-points. Note that this approach requires the assumption that $m/z$ 60 should be predominantly explained by BBOA, which is likely true when the fraction of OA signal occurring at $m/z$ 60 ($f$60) is relatively high. However, for measurements where $f$60 is low, $m/z$ 60 is more likely to have also contributions from other sources. A rough guideline for utilizing this criterion is a threshold for biomass burning

influence of $f$60 = 0.003 as identified by Cubison et al. (2011). In the current dataset, ~85 % of all measured time-points exceeded this threshold. Every measured day was observed to comprise at least some time-points (in winter, spring and fall almost all points whereas in summer mostly evening points) above this threshold, suggesting that the criterion is valid throughout the dataset.



Ng et al. (2010) described higher $f43$ and lower $f44$ for the mass spectrum of SV-OOA, and *vice versa* for LV-OOA. Therefore,

355 $f43$ and $f44$ are used as proxies for SV-OOA and LV-OOA or OOA, respectively. For LV-OOA (Figure 2 d, Table 2) all score values are allowed here, whereas for SV-OOA (Figure 2e, Table 2) the PMF runs meeting the thresholds for the five-factor solutions are selected. This threshold corresponds to the point where $n_{\text{non-modelled}}$ is minimal with respect to this criterion, i.e. considering all PMF runs in this criterion leads to the same $n_{\text{non-modelled}}$, at highest possible $f43$ for SV-OOA.

The criterion of SV-OOA is further used to differentiate between four- and five-factor solutions on the window-runs. For the

360 PMF windows where no five-factor solution with SV-OOA is selected, the set of four-factor solutions in the corresponding PMF window is automatically selected (green points at zero in Figure 2e). Finally, the averaging procedure also controls and prevents that four- and five-factor solutions are simultaneously considered for the averaging of single time-points by privileging five-factor solutions, i.e. any time-point containing accepted PMF runs with both 4- and 5-factor solutions retains only the 5-factor solution.

## 365 3 Results

### 3.1 Brief statistical analysis of the rolling result

The amount of $n_{\text{non-modelled}}$ resulting from the criteria and thresholds reported in Table 2 yields 99.31 % data coverage, corresponding to a total of only 3 non-modelled days. Overall, the selected criteria resulted in 1'970 accepted PMF runs (~5.6 % out of the 35'100 PMF runs). The $Q/Q_{exp}$ has an average value of 4.4, a median of 4.8, and the first and third quartiles are

370 3.7 and 5.5, respectively. These values are reasonable, given that many previously conducted AMS studies reported values between 1 and 10 (Zhang et al., 2011). On average, each data point has 43 replicates (median = 24, first and third quartiles 9 and 60, respectively), which are used to assess the statistical uncertainty of the PMF solution as discussed in Section 0.

### 3.2 Factor time series

### 3.2.1 Overview

375 Figure 3a shows the time series of each factor for the entire dataset as a mean, averaged over all accepted PMF runs. The data from Figure 3a is re-averaged to monthly and seasonal means and shown in Figure 3b and 3c, respectively. For Figure 3c, seasons are defined as follows: winter is December - February, spring is March - May, summer is June - August, and fall is September - November.

In winter, spring and fall the concentrations of primary organic aerosols (HOA, COA and BBOA) are approximately 40 %

380 compared to the 60 % of the (secondary) oxygenated organic aerosols (SV-OOA, LV-OOA or OOA). In summer the primary fraction decreases to reach minimum values of 30 % compared to 70 % of OOA. The relative fractions of HOA and COA are rather constant, contributing on average between 0.4-0.7 $\mu$g·m$^{-3}$ (7.8-9.0 %) and 0.7-1.2 $\mu$g·m$^{-3}$ (12.2-15.7 %), respectively throughout the year. BBOA shows a strong yearly cycle with the lowest mean concentrations in summer (0.6 $\mu$g·m$^{-3}$, 12.0 %),



slightly higher mean concentrations during spring and fall (1.0 and 1.5 µg·m⁻³, or 15.6 and 18.6 %, respectively) and highest

mean concentrations during winter (1.9 µg·m⁻³, 25.0 %). Only during summer, the bulk OOA is completely separated into SV-OOA and LV-OOA, with mean concentrations of 1.4 µg·m⁻³ (26.5 %) and 2.2 µg·m⁻³ (40.3 %), respectively.

For the remaining seasons the seasonal concentrations of SV-OOA, LV-OOA and OOA comprise 0.3-1.1 µg·m⁻³ (3.4-15.9 %), 0.6-2.2 µg·m⁻³ (7.7-33.7 %) and 0.9-3.1 µg·m⁻³ (13.7-39.9 %), respectively.

The time series of the primary OA factors HOA, COA and to some extent BBOA are rather spiky (Figure 3a), underlining a strong influence of local sources. The COA spikes that are present from May 2011 through the end of September 2011 are likely due to local barbecuing events during the evening, as also observed in an earlier study at this site (Lanz et al., 2007). The highest COA concentrations are observed in early July 2011, where the NR-PM₁ mass concentrations reached 70 µg·m⁻³, and correspond to three consecutive evenings/nights of a yearly Latin American dance and grill festival (Caliente). During this

festival, the courtyard containing the measurement site was filled with food and grill stands, explaining the dominant contribution of COA. Throughout the summer and spring and less frequently in autumn/winter SV-OOA was modelled in addition to LV-OOA. This warm period was characterized by high daily temperatures and induced on the one hand variability in the condensed OOA allowing for separation of SV-OOA and LV-OOA and on the other hand increased emissions of biogenic SV-OOA precursors (Canonaco et al., 2015).

### 3.2.2 Daily cycles

Figure 4 summarizes the weekday (left) and weekend (right) daily cycles for the modelled factors. The daily cycle of HOA follows the averaged daily cycles of the estimated traffic of eBC (eBC$_{tr}$) and of NO$_x$. The same is true for the daily cycle of BBOA following that of the biomass burning of eBC (eBC$_{wb}$). HOA, eBC$_{tr}$ and NO$_x$ exhibit a clear rush-hour peak on weekdays and none on the weekend. During the weekdays, a small lunch peak is visible for COA underlying the meal activity during the

working days and the presence of many restaurants in this area. There are no evident differences between the weekday and weekend daily cycles of LV-OOA, SV-OOA and OOA. LV-OOA and OOA show rather flat daily cycles, similar to their inorganic aerosol tracers SO$_4^{2-}$ and NH$_4^+$, respectively. This is in line with their most-likely regional background, as already suggested earlier (Canonaco et al., 2015). Only the concentration of SV-OOA tends to decrease during the afternoon, suggesting its volatile nature, similarly to its inorganic aerosol tracer NO$_3^-$. The weekly cycle for HOA, COA, BBOA and the

OOAs including their tracers eBC$_{tr}$, NO$_X$, eBC$_{wb}$, SO$_4^{2-}$, NO$_3^-$ and NH$_4^+$, respectively are reported in Supplement B. Apart from OOA, the weekly cycle for HOA, BBOA, SV-OOA and LV-OOA are in good agreement with their tracers.

### 3.2.3 Comparison with external data

The analysis and further validation of the PMF runs using the criteria-based selection are performed on the PMF results of the rolling windows and therefore, correlations are performed over 14 days in this study. The performance of the rolling strategy





can then be verified by the factor / tracer correlation, e.g., on average over the seasons (Table 3). Moreover, the same factor to tracer correlations are also evaluated for the seasonal *pre*-tests (PMF runs over the seasons with no rolling strategy) and are reported in brackets in Table 3.

$NO_x$ data is available only in winter and fall 2011. Both $NO_x$ and $eBC_{tr}$ are correlated with HOA over the full year and within individual seasons. The correlation values with $NO_x$, are lower compared to those found in Canonaco et al. (2013). However,

in Canonaco et al. (2013) the data covered mostly the two winters including some parts of spring and fall. For the latter two seasons $NO_x$ data was not properly validated and was consequently removed from further analysis (no $NO_x$ data is available for spring and summer). Moreover, in Canonaco et al. (2013) the model validation was strongly based on the first winter period, and when performing the correlation between HOA and $NO_x$ data for that period only, the correlations were similar also in the current study (not shown in the table).

BBOA shows substantial correlation to $eBC_{wb}$ in fall and winter, as also found in Canonaco et al. (2013), while the correlation is low in spring and very low in summer. These low correlations are expected, since the determination of $eBC_{wb}$ is highly uncertain when the $eBC_{wb}/eBC_{traffic}$ ratio is low. Wood burning source apportionment of eBC data, as already stressed above, is not suited under warm conditions with low biomass burning contributions. However, the correlation is good over the full year, as the problematic data anyways yields $eBC_{wb}$ concentrations near zero, and the correlation is thus driven by the data

with high signal to noise ratios.

High correlations between LV-OOA and $SO_4^{2-}$ are seen over the year as well as for spring and fall, whereas they are lower in summer, as shown in Table 3, in contrast to Lanz et al. (2007) ($R_{Pearson} = 0.5$ between LV-OOA and $SO_4^{2-}$ during a summer AMS campaign). The correlation between SV-OOA and $NO_3^-$ is higher for winter 2011 and summer but lower in spring and fall. This is understandable, as the spring and fall represent the transition between modelling SV-OOA and LV-OOA (summer)

compared to one OOA only (winter). The correlation between SV-OOA and LV-OOA for winter 2012 is not shown due to the low number of time-points for which both OOAs were modelled. OOA correlates well with $NH_4^+$ throughout the year in accordance to summer and winter data reported previously (Lanz et al., 2007; Lanz et al., 2008; Canonaco et al., 2013). In contrast to the OOAs, few differences are observed for BBOA, HOA, or COA between the two winters. This supports the conclusion that the different OOA behavior in these two winters reflects actual meteorological and chemical differences rather

than mixing and / or splitting between the POA and SOA factors.

Importantly, the rolling results show generally higher correlations with the external tracers than do the conventional seasonal PMF runs (values in brackets in Table 3). This demonstrates that the rolling approach generally outperforms the conventional seasonal PMF analysis.

### 3.3 Time-dependent factor profiles

The mean factor profiles of the six modelled sources/components over the entire year are presented in Figure 5. Error bars show one standard deviation of profile variability across the entire measurement year. Note that this variability comprises both

the time-dependent variation of the factor profiles and the PMF error (see Section 0. for more details on the discussion of the errors in this study).

A better understanding of the temporal variation of the factor profiles is gained when inspecting them over time. Figure 6

shows the fractional contributions of $m/z$ 41, 43, 44, 55, 57 and 60 to each factor profile as a function of time. Each variable is normalized by its mean contribution. In general, the variation of the fractions for the primary OA factors (HOA, COA and BBOA) seems small compared to the variability of the oxygenated factors (LV-OOA, SV-OOA and OOA). The primary OA factors show low profile variability with almost no seasonal pattern. Note that minimum and maximum values of these variables for the primary OA factors (less pronounced for HOA and COA) reach ~0.6 and 1.4, respectively, i.e., the boundaries

given by $a_{max}$. The 75$^{th}$ percentiles of the $a$-values for HOA, COA and BBOA touches $a_{max}$ less than 0.9 % of the time and the 90$^{th}$ percentile hits $a_{max}$ 34 %, 24 % and 73 % of the time (see Supplement D Figure S5). This suggests that the factor profiles are not limited by the constraining technique, but rather by the employed scheme of criteria. Allowing for higher $a_{max}$ and loosening the criteria threshold would most likely increase the variability in these ions but would also lead to mixed and environmentally unreasonable solutions.

This is different for the oxygenated factors. LV-OOA, SV-OOA and OOA for example contain high $m/z$ 60 for the colder season, likely indicating biomass burning influences (Canonaco et al., 2015; Heringa et al., 2011, Qi et al., 2019). In addition, $m/z$ 57 shows a strong seasonal pattern, i.e., high in winter and low during summer for SV-OOA and LV-OOA. Strong peaks are also observed for $m/z$ 43 in LV-OOA during summer. This is due to less oxygenated bulk LV-OOA compared to the winter in Zurich, when LV-OOA or OOA represent more oxygenated aerosol with higher $m/z$ 44 and lower $m/z$ 43, as already noted

in Canonaco et al. (2015). SV-OOA also contains a very strong increase in $m/z$ 55 during the Caliente episode. Most likely one COA factor alone is insufficient to capture all the variability of $m/z$ 55. As a consequence, PMF uses an additional factor for modelling the variability of $m/z$ 55, here SV-OOA which may contain some characteristics of cooking SOA, as the latter has been shown to have non-negligible contribution at $m/z$ 55 as well (Klein et al., 2016). Further evidence comes from Figure 6e (and also Supplement C Figure S4), where $m/z$ 55 and $m/z$ 43 peak around Caliente in SV-OOA and LV-OOA, respectively.

Moreover, $m/z$ 44 drops in LV-OOA. This implies that SV-OOA has some characteristics of cooking while LV-OOA becomes more SV-OOA-like during Caliente. The period of influence of these peaks lasts until 8-10 days before and after Caliente, most likely as it is incorporated during the window-runs 14 days before and after Caliente.

The time-dependent mass spectral matrix of the factors can be found in the Supplement, section C, although a detailed analysis is beyond the scope of the current study. When employing this type of analysis, future studies should investigate in more detail

changes of the variables in the factor profiles. This information might provide new insights on seasonal or source-specific markers, essential for source apportionment analyses.

### 3.4 Residual analysis

Figure 7a and b show the scaled residuals as functions of $m/z$ and time, respectively. The scaled residuals do not reveal any systematic over- or underestimation. The data scatters around zero with the interquartile range almost always between +/- 3





throughout the entire year evidencing the good quality of the PMF solution on average (+/- 3 is the reasonable range for scaled residuals defined in Paatero and Hopke (2003)). The highest residuals occur during the Caliente festival (beginning of July), as shown by the dark red spike (interquartile range) in the time series plot (Figure 7b), when the PMF solution is strongly influenced by extremely local and short-term cooking and biomass burning sources that are not fully captured by the retrieved COA and BBOA factors.

This results in a change of the factor profiles of COA and BBOA and SV-OOA (as already stressed in Section 0). However, the COA, BBOA and SV-OOA profiles roughly 8-10 days before and after Caliente are again consistent with those retrieved during the rest of the season, i.e. the unique fingerprint during the Caliente episode does not strongly influence the solution of the PMF-windows around Caliente. A few other episodes in spring (May) and at the end of the summer (September) reach also higher scaled residuals. In the current dataset, these likely indicate PMF runs that have not fully captured profile responses

to rapid meteorological changes (colder to warmer season and *vice versa*). This happens on a shorter time scale than the chosen PMF window and as a consequence cannot be fully captured by the 14-day PMF windows, causing PMF solutions with mixed factor profiles and higher scaled residuals. Note that during the last third of the campaign the scaled residual distribution tends to be broader. This is due to technical problems on the ACSM inlet system mainly related to the filter valve clogging, causing noisier signals and consequently noisier PMF results for the valve switching system employed at that time. This condition is

not accounted for by the ACSM error model and increases the scaled residuals.

### 3.5 Uncertainty of the PMF solution

Within this study, each PMF run combines a random selection of *a*-values for the three constrained POA factors with random (time-based) resampling of the input matrix. PMF runs satisfying the acceptance criteria are retained for the final result leading to several repeats for each time-point *i*. The variability among these repeats at each *i* can be used to infer the rotational and

statistical uncertainty. These two types of uncertainties are discussed below and are collectively referred to as PMF error within this study. Additional contributions to the overall uncertainty of this analysis that are not assessed here include anchor profile selection, as well as the error related to the criteria construction, such as the type of criterion (correlation, diurnal, profile characteristics, etc.), tracer selection, and its related threshold selection. The proposed relative PMF error in percentage in this study is given by the following formula:

$$PMF_{error} = \frac{100}{2 \cdot n} \cdot \sum_{i=0}^{n} \left(\frac{\sigma}{avg}\right)_i \tag{6}$$

where $\sigma$ is the standard deviation and avg is the mean value of all replicates of a time-point *i*. The probability density function (pdf) of $PMF_{error}$ for each time-point $i \left(\frac{\sigma}{avg}\right)_i$ is reported in Figure 8. The relative PMF errors are given by the center of the lognormal fit ($x_0$) as visualized in Figure 8 and are for HOA, COA, BBOA, LV-OOA, SV-OOA and OOA ± 34 %, ± 27 %, ±30, ±11 %, ±25 % and ±12 %, respectively.





The data reported in Figure 8 was first log-transformed, as the untransformed distribution was skewed to the right, mostly due to time-points with low signal to noise ratio that would have had a stronger impact on the final error calculation using an untransformed, i.e., linear representation.

## 4 Recommendations and current limitations

The techniques described in this study are relevant for long-term source apportionment (SA) studies, in particular for ACSM
data. The stability of the primary profiles (HOA, COA and BBOA) suggests that they are rather independent from the season, and that employing primary OA factors coming from other SA studies (here profiles from an AMS SA in Paris conducted years earlier) using, e.g., the $a$-value constraints, works even for long-term SA. However, this outcome is not completely independent as it results from the defined $a_{max}$ as well as the applied scheme of criteria with their corresponding criteria thresholds. Increasing these thresholds would most likely increase the variation in the POA factor profiles but would also favor
more mixing between these factors. Significant seasonal changes in factor profiles were found for SV-OOA and LV-OOA. Hence, the rolling mechanism is essential, when accurately apportioning the oxygenated organic aerosol fraction. The use of a 14-day window, as already proposed by two former studies (Fröhlich et al., 2015; Parworth et al., 2015), was shown to be appropriate for this long-term SA analysis and represents a promising starting point for future long-term SA studies, although detailed evaluation for datasets with other sources and temporal characteristics is needed.

In general, selection of the rolling window size ($s_{win}$) should consider both the fraction of non-modelled time-points (see Figure 1) and interactions between $s_{win}$ and solution acceptance criteria. The latter point is illustrated by the use of the relative intensity of the COA lunchtime peak in this study. This peak was observed to be almost absent during the weekend. As a consequence, avoiding systematic biases in the fraction of non-modelled time-points requires the $s_{win}$ to be larger than 7 days to guarantee the presence of weekdays in every window-run. Employing a reliable tracer even during the weekends for the cooking source
would have allowed for a better exploration of $s_{win}$ below 7 days, as similar $Q/Q_{exp}$ values resulted for 3, 7 and 4 days windows, as shown in Figure 1.

The importance of defining the proper number of factors is strongly emphasized when analyzing transient events, e.g. the Caliente episode. This becomes even more important when performing automated source apportionment schemes, where the ability of factors to dynamically change and adapt to the current window-run is limited, as it is the case for the current rolling
mechanism presented in this study. During Caliente the variability of $m/z$ 55 required two cooking factors to be fully described. With only one cooking factor allowed, other unconstrained factors (especially SV-OOA) took on some cooking characteristics. This resulted in mixed SV-OOA and LV-OOA factors, as $m/z$ 55 and $m/z$ 43 were clearly peaking around Caliente for SV-OOA and LV-OOA, respectively. Relevant transient events that should still be part of the SA result would most likely require further attention with additional and separate PMF runs, where the user can better control the required number of factors and
$s_{win}$. Such problems are clearly evident from diagnostics such as increased residuals (Figure 7b) and sudden changes in factor profiles (Figure S3 and S4), facilitating their appropriate identification and treatment. A 14-day window is likely too large for



transient events representing a small fraction of $s_{win}$, where the latter strongly influences the contributions of the data for $s_{win}$ days around the event.

Crippa et al (2014) already demonstrated for 25 AMS datasets that an $a_{max}$ of 0.3 for the constrained information was often

required for those SA studies. For the present algorithm and dataset, an $a_{max}$ of 0.4 was shown to be ideal. Smaller $a_{max}$ did not allow the constrained profiles to sufficiently adapt to the data, whereas higher values were subject to mixing of the profiles. $a$-value limits strongly depend on how well the fingerprint matches the PMF input. Fingerprints applied obtained by SA analyses of other locations or during other meteorological conditions might require a higher $a$-value limit compared to those extracted from, e.g., a *pre*-analysis conducted on a subset of the PMF input.

The other remaining free parameters ($x_{PMF}$ and in particular the choice of the criteria and their corresponding thresholds) must be assessed by the user for any new SA study, as they may strongly depend on site/source characteristics and tracer availability. Moreover, investigation of various tracers as criteria-candidates for one source is also very desirable, as it allows to quantify errors when discussing factor-tracer interchangeability.

Unlike batch-style PMF (i.e., a single PMF run encompassing the entire dataset), here corrections or scaling factors affecting

entire rows or columns of the input data matrix should be applied prior to SA analysis. For example, the collection efficiency (CE) parameter applied for ACSM data analysis is applied to all measured $m/z$'s of a mass spectrum and does not alter the relative contributions obtained by a single PMF result. However, it does affect the overall source apportionment returned by the rolling window strategy presented within this study. This comes from the fact that the final source apportionment result is the aggregate of a set of accepted solutions whose criteria for acceptance may include goodness of correlation with an external

tracer, and such correlations are affected by CE. Therefore, applying CE post-PMF will require the user to re-evaluate the score plots and to reassess the criteria thresholds.

It is likely that the PMF errors reported above can be further reduced by further refinements to the rolling window algorithm. One major limitation is the application of season-specific criteria thresholds. In the future, criteria thresholds with a higher temporal resolution are certainly desirable. Another major limitation is the continuous presence of the primary OA factors

during the entire analysis. Similarly to the (de)activation of SV-OOA within this study, in the future one or more factors should be (de)activated during the evolution of the rolling approach to better cope with the complex and dynamic real atmospheric conditions.

## 5 Conclusion

A rolling-window PMF algorithm was applied to NR-PM$_1$ organic data measured with an ACSM between February 2011 and

February 2012 in downtown Zurich, Switzerland. The rolling approach allows for a source apportionment of time-dependent factor profiles and has several advantages, e.g., very fast PMF runs of rather small PMF runs (few seconds for 14 days windows) compared to conventional batch analysis (several minutes, as PMF run is always the entire dataset) or one factor per source compared to several factors in batch analysis to cope with time-varying factor profiles. Moreover, the rolling technique



is particularly helpful for the analysis of automated and / or continuous analysis of both long-term and continuously growing

datasets, where batch analysis is at best inefficient and probably not feasible. Factor/tracer correlations were shown to be higher for the averaged seasonal analysis (from the rolling window) than for the seasonal *pre*-tests (PMF runs with no rolling). This highlights the improved performance of the rolling PMF runs compared to conventional batch PMF analysis for long-term data.

PMF runs were conducted where the *a*-values of the constrained factor profiles were randomly changed within the boundaries

0 to $a_{max}$ in conjunction with the bootstrap resampling strategy. The resulting PMF runs were selected and studied using the criteria scheme based on information on the sampling site from previous SA studies. This method has shown its usefulness when evaluating and studying hundreds of thousands of PMF runs. The criteria used here consisted of features in the diurnal patterns of HOA and COA, the amount of explained variation of *m/z* 60 attributed to BBOA, and representation of OOA by one or two factors depending on the difference between SV-OOA and LV-OOA in *f*43 values.

The separation between the primary OA factors (HOA, COA and BBOA) and oxygenated organic aerosol (SV-OOA, LV-OOA and OOA) was rather robust throughout the year. HOA and COA were rather constant, whereas BBOA showed a very strong seasonality with the highest contribution in winter and lowest in summer. The model separated OOA into SV-OOA and LV-OOA mainly during the warm season (spring and summer), including a warm episode during the first winter. Strongest changes of the factor profiles where visible for the oxygenated species SV-OOA and LV-OOA, whereas the primary species

HOA, COA and BBOA showed smaller variations. Hence, the rolling mechanism is certainly essential when properly apportioning the oxygenated organic aerosol fraction.

The model was still able to separate a semi-volatile fraction for the colder seasons based on the variation in *m/z* 43 and 44, where very little variation was present in nitrate, often used as a tracer of SV-OOA.

The rotational and statistical uncertainties were assessed via random *a*-values exploration and bootstrap resampling. The

relative PMF errors (expressed by the standard deviation divided by the average concentration of all replicates per time-point) are on average ± 34 %, ± 27 %, ±30, ±11 %, ±25 % and ±12 % for HOA, COA, BBOA, LV-OOA, SV-OOA and OOA, respectively.

Finally, the free parameters tested and validated in this study, i.e., the 14-day window length, 0.4 as upper limit for the *a*-value of the constrained primary OA factor profiles, together with the scheme of criteria and the $x_{PMF}$ per window-run, depend on

the sources and meteorological conditions of Zurich downtown. When applying this new rolling strategy on datasets dissimilar to Zurich, some or all of these parameters might be subject to investigation to achieve a complete and quantitative source apportionment analysis.

## 6 Acknowledgements

We would like to thank the COST action CA16109 Chemical On-Line cOmpoSition and Source Apportionment of fine

aerosoLs (COLOSSAL) and the SNF COST project SAMSAM IZCOZO_177063 for supporting part of the work described



herein. F. Canonaco and C. Bozzetti have also been / are still employed by Datalystica Ltd. during the final development of the main SoFi Pro packages and Datalystica Ltd. is the official distributor of the SoFi Pro licenses.

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

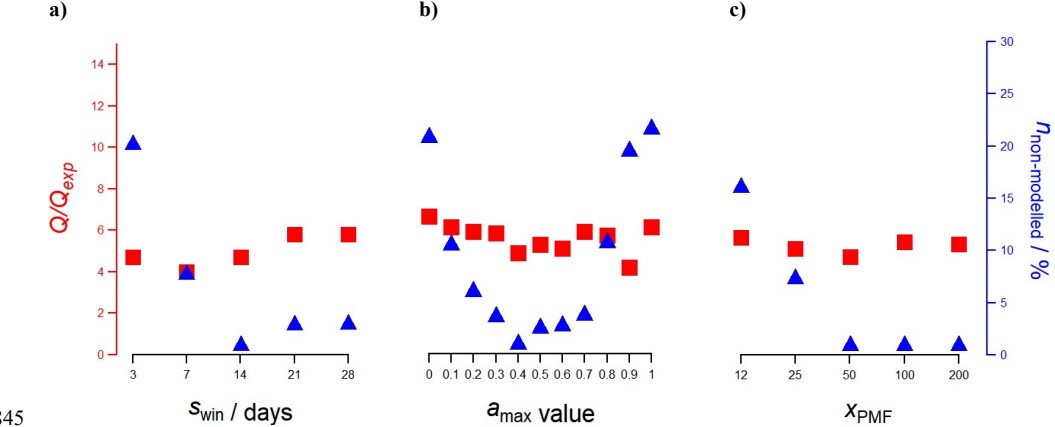


**Figure 1: The mathematical metric Q/Qexp (left axis, red points) and the percent of non-modelled time-points (nnon-modelled) (right axis, blue points) over the entire dataset are reported as a function of window size (swin), maximum a-value (amax), and number of PMF repeats per window (xPMF). In each plot, two of these three parameters are fixed at their optimum values and the third is varied: (a) swin, (b) amax, (c) xPMF. Optimum values are swin = 14 days, amax = 0.4, and xPMF = 50. For all runs, criteria**
**are defined as described in Section 0.**

**Table 1: Overview of the rolling mechanism and the repeats of the PMF analysis.**





| Rolling mechanism |
| --- |
| ➢ a 14-day time window is defined |
| ➢ window is shifted by one day over the entire dataset |
| **PMF analysis** |
| ➢ for each window a four- and five- (HOA, COA, BBOA and one up to two OOAs) factor PMF run is performed, where HOA, COA and BBOA are constrained within the *a*-value approach. |
| ➢ PMF runs are initialized 50 times from random starting points for the unconstrained information in **G** and **F** (seeds). The *a*-values for the constrained factor profiles are randomly and independently varied from $a = 0$ to $a = 0.4$ with a resolution of $\Box a = 0.1$ (*a*-value exploration). In each run the PMF input is resampled within the bootstrap method. |


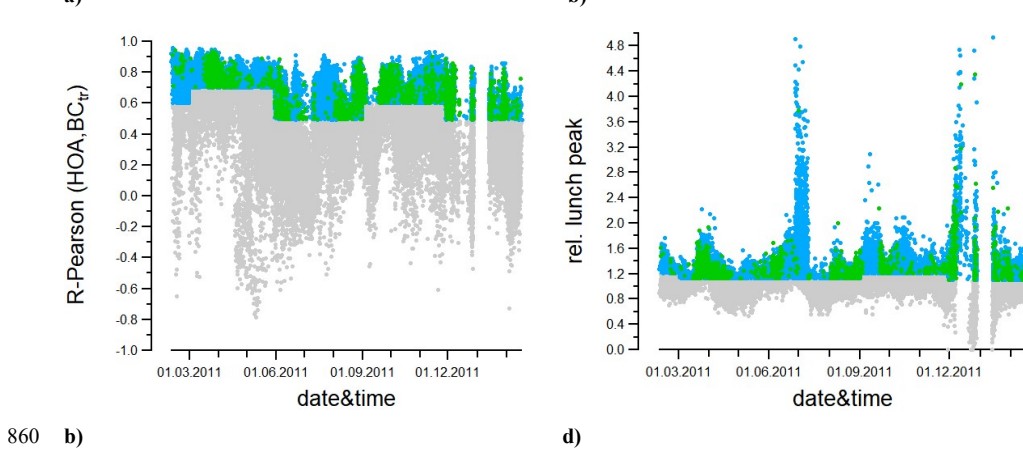

**b)**                                                    **d)**

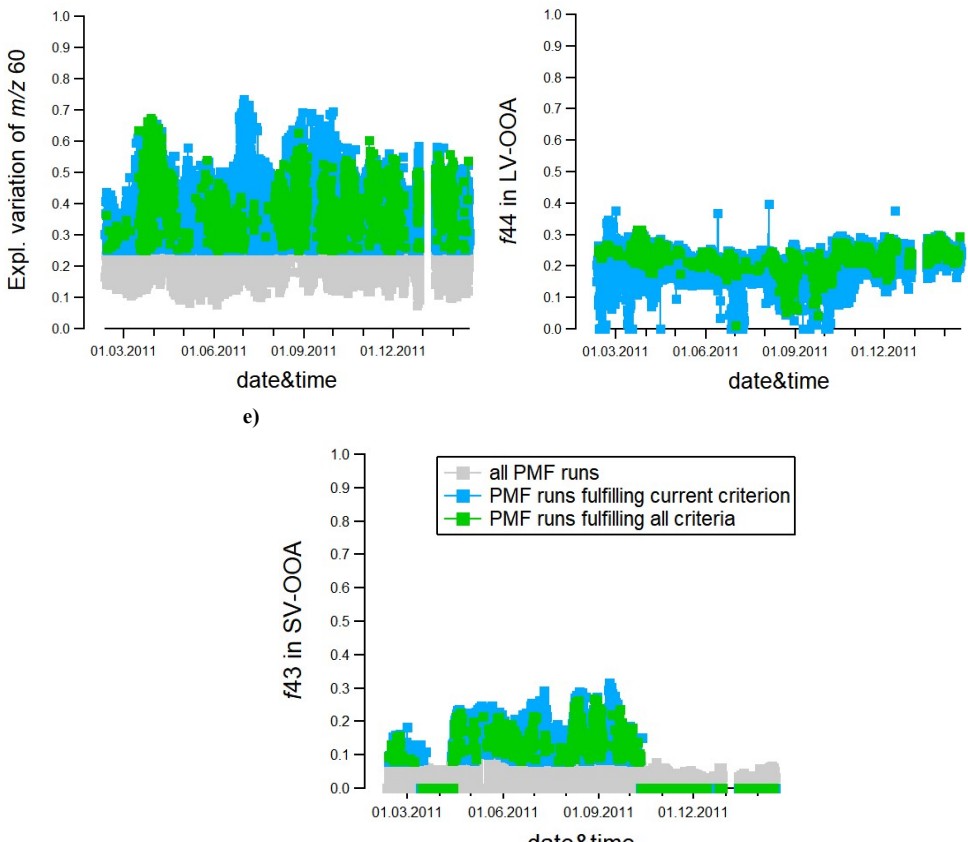

**Figure 2: PMF runs sorted based on the scores (gray points), PMF runs fulfilling the criterion thresholds (blue points) and PMF runs fulfilling criterion thresholds in all criteria (green points). The five criteria are a) diurnal correlation between HOA and eBCtr (seasonal thresholds from statistical analysis), b) relative lunch peak for COA (seasonal thresholds from statistical analysis), c) explained variation of m/z 60 for BBOA, d) f44 in LV-OOA profile and e) f43 in SV-OOA profile, respectively. Note that e) contains three episodes with zero points, which represent four-factor solutions automatically selected by the algorithm, where no five-factor solution was manually selected (and the SV-OOA criterion is thus irrelevant).**

**Table 2: Criteria scheme employed in this study. The first value represents the applied threshold for the final PMF solution and the values in brackets for HOA and COA stand for the threshold value coming from the seasonal resampling analysis. f44 for LV-OOA is used for factor sorting rather than as an acceptance/rejection threshold.**

| factor | criteria types | criteria thresholds |
|---|---|---|





|  |  | winter 2011 | spring 11 | summer 2011 | fall 11 | winter 2012 |
|---|---|---|---|---|---|---|
| HOA | daily cycle correlation ($R_{Pearson}$) between HOA and eBC$_{traffic}$ | 0.6 (0.6) | 0.7 (0.8) | 0.5 (0.2) | 0.6 (0.6) | 0.5 (0.2) |
| COA | rel. lunch peak (11+12+13 hrs) to (9+10+14+15 hrs) | 1.2 (1.2) | 1.1 (1.1) | 1.1 (1.1) | 1.2 (1.2) | 1.1 (1.1) |
| BBOA | explained variation of $m/z$ 60 | 0.25 | 0.25 | 0.25 | 0.25 | 0.25 |
| LV-OOA | 44 in profile | N/A | | | | |
| SV-OOA | $f$43 in profile | 0.08 | | | | |

**a)**

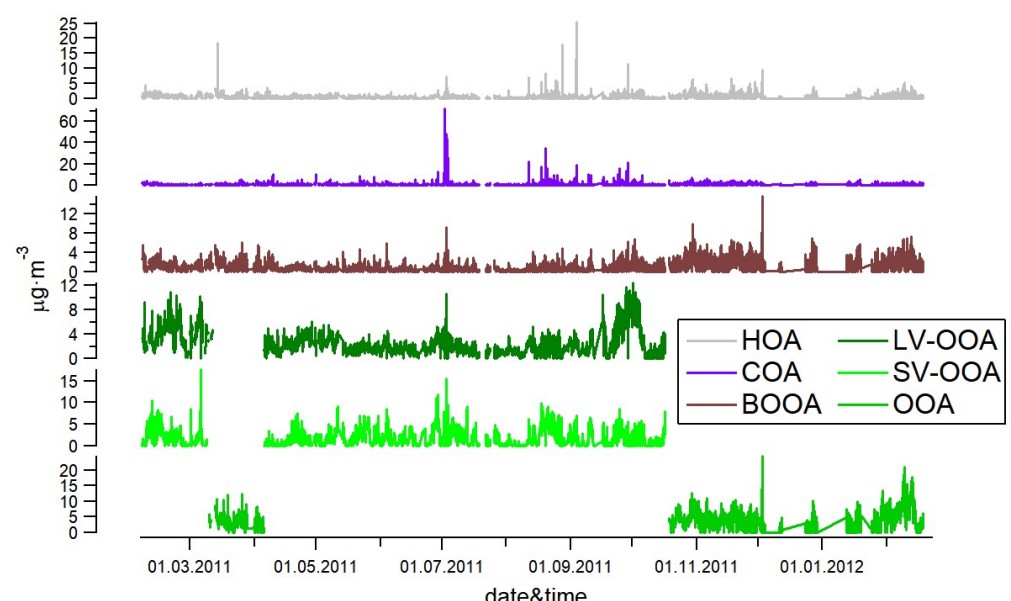





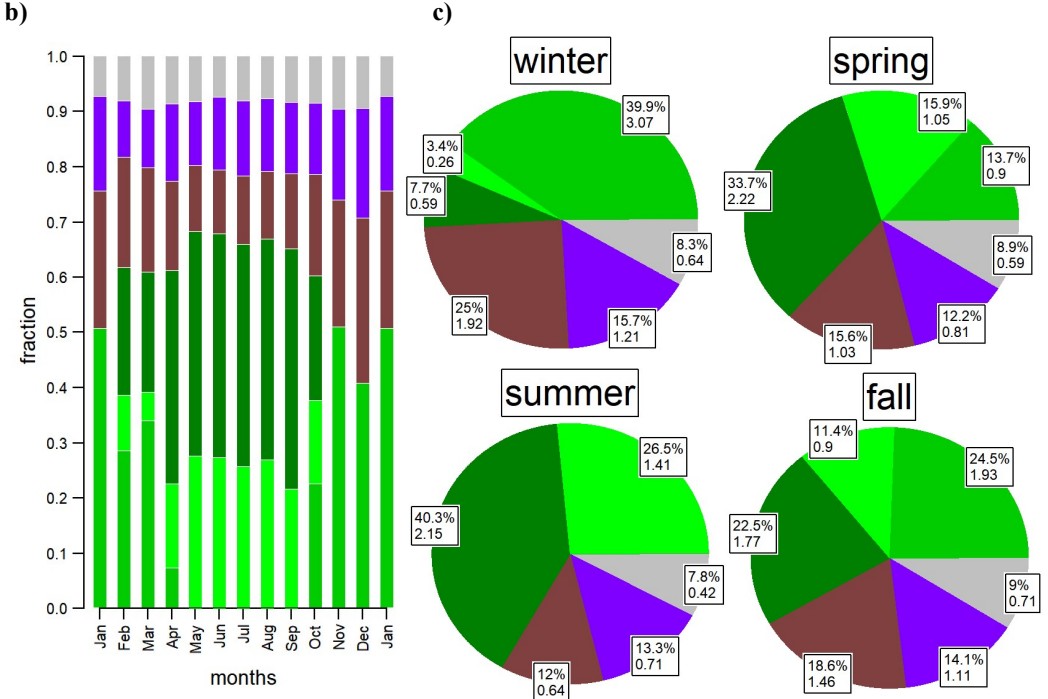

Figure 3: a) 30 minutes average concentrations b) relative contributions c) pie charts for the calendar seasons of the sources between February 2011 and February 2012. Gaps in the data represent interruptions due to maintenance and / or technical problems of the ACSM during the last third of the campaign, mostly due to clogging issues on the ACSM inlet. The lower values in the pie charts are the seasonal mean contributions in μg·m-3. Note that the OOA factors are represented either as LV-OOA and SV-OOA (5-factor solution) or OOA alone (4-factor solution).





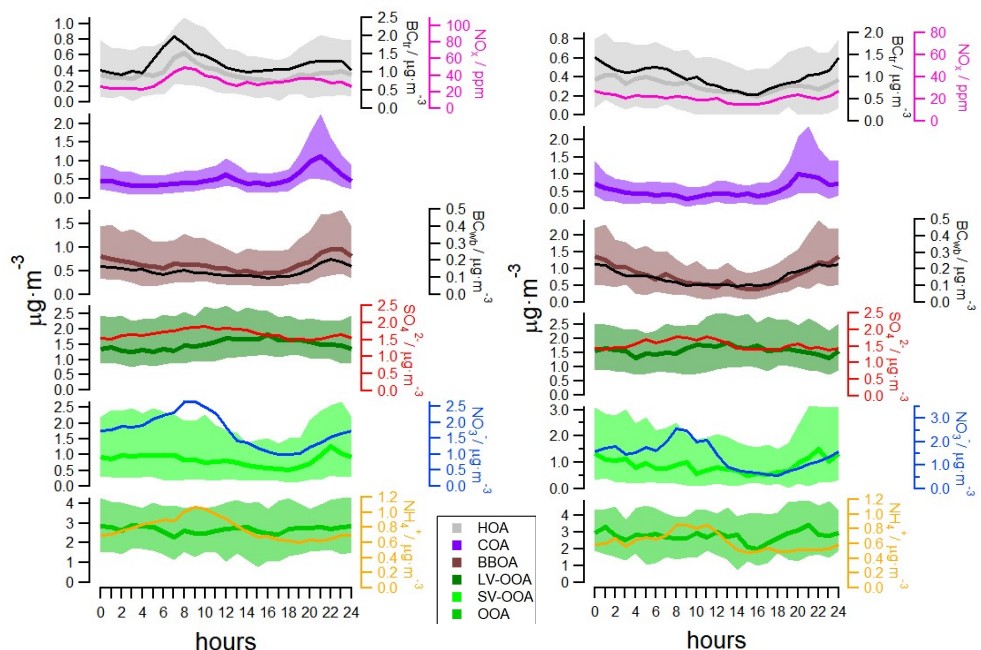

**Figure 4:** The weekday (left) and weekend (right) diurnal cycles for the entire period (February 2011 – February 2012).
The thick lines represent the medians and the shaded areas span the interquartile ranges. Typical external tracers are also shown for comparison, i.e., $eBC_{tr}$ and $NO_x$ for HOA, $eBC_{wb}$ for BBOA, $SO_4^{2-}$ for LV-OOA, $NO_3^-$ for SV-OOA and $NH_4^+$ for OOA.


**Table 3:** Correlation coefficients ($R_{Pearson}^2$) with a significance level of $p >= 0.01$ between the factor contribution and expected tracers over the year and the meteorological seasons as defined above. The first value describes the correlation for the rolling result, whereas the value in brackets is for the seasonal PMF result (no rolling).

| factor | year | winter 2011 | spring 2011 | summer 2011 | fall 2011 | winter 2012 |
|---|---|---|---|---|---|---|
| HOA / $NO_x$ | 0.29 | 0.18 (0.21) | - | - | 0.33 (0.24) | 0.17 (0.18) |
| HOA / $eBC_{tr}$ | 0.36 | 0.45 (0.44) | 0.28 (0.28) | 0.22 (0.08) | 0.38 (0.31) | 0.42 (0.27) |
| COA | - | - | - | - | - | - |
| BBOA / $eBC_{wb}$ | 0.32 | 0.36 (0.23) | 0.22 (0.07) | 0.06 (0.01) | 0.35 (0.22) | 0.43 (0.41) |
| LV-OOA / $SO_4^{2-}$ | 0.48 | 0.37 (0.41) | 0.60 (0.50) | 0.30 (0.26) | 0.54 (0.30) | - |





| SV-OOA / NO$_3^-$ | 0.05 | 0.24 (0.06) | 0.03 (0.01) | 0.31 (0.29) | 0.15 (0.04) | - |
| OOA / NH$_4^+$ | 0.60 | 0.71 | 0.58 | - | 0.39 | 0.70 (0.59) |


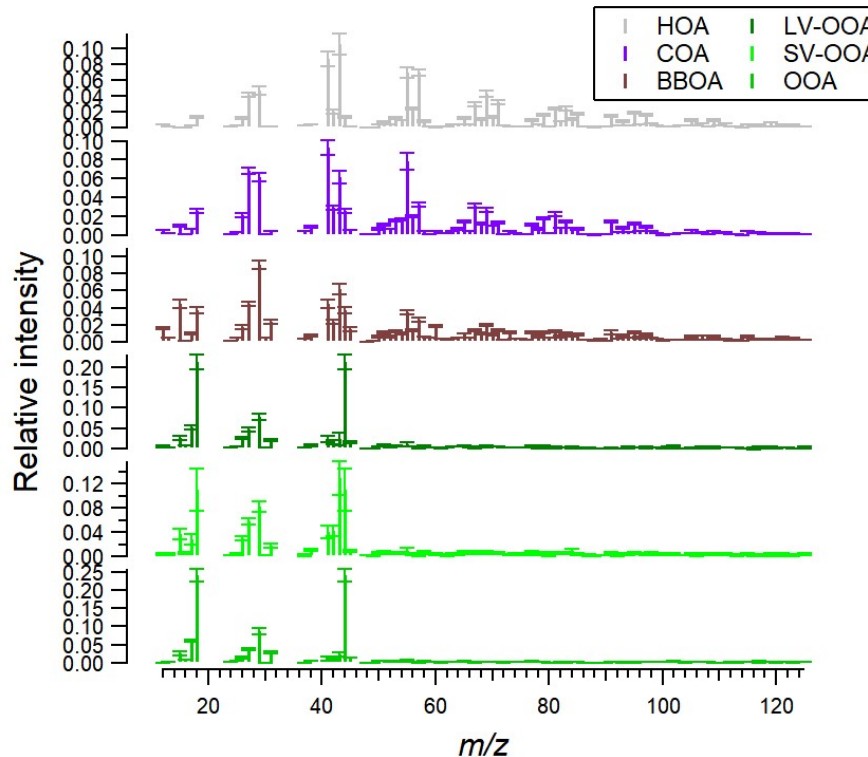

**Figure 5: The mass spectra of the six factors. The spectra have been truncated at *m/z* 100 to facilitate the comparison of the key *m/z* in the lower range. Error bars represent one standard deviation of the profile variability across the entire**
**year.**


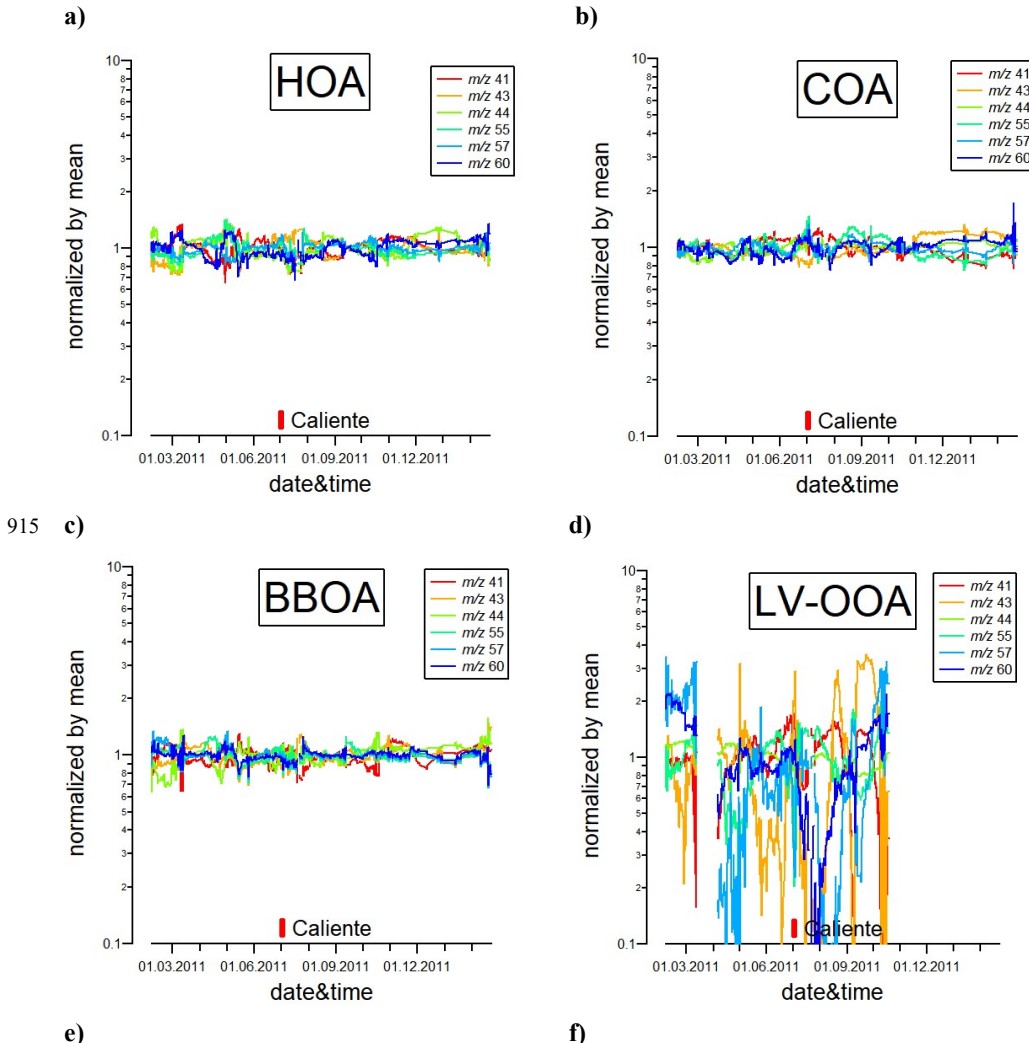






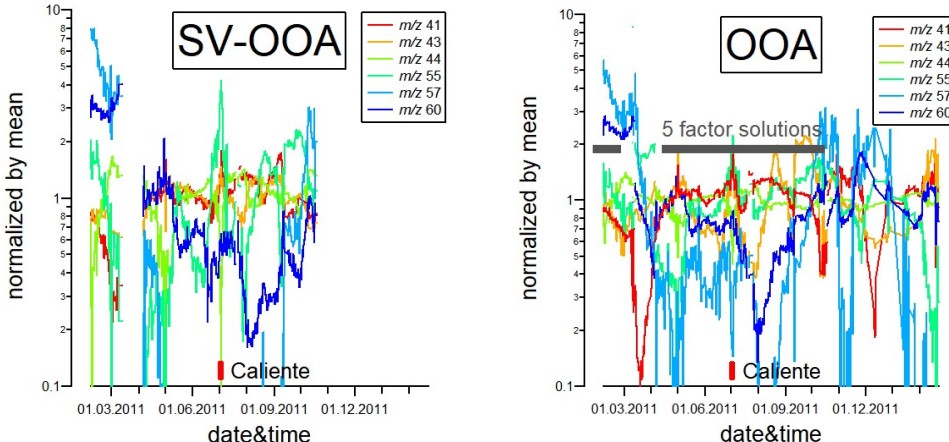

**Figure 6: Daily averaged fractions of important AMS/ACSM variables. Each variable is normalized by its mean to better stress its temporal variation.**


a)                                                    c)






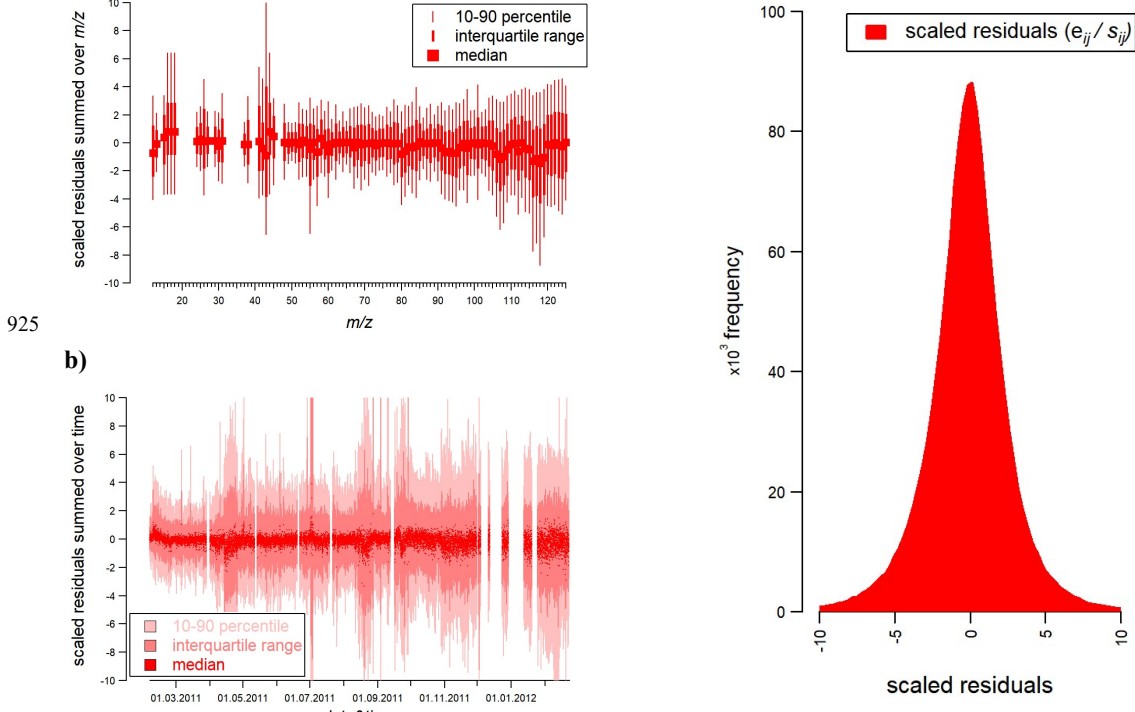

**Figure 7: a) scaled residuals over *m/z*'s, b) scaled residuals over time and c) total histogram of scaled residuals.**




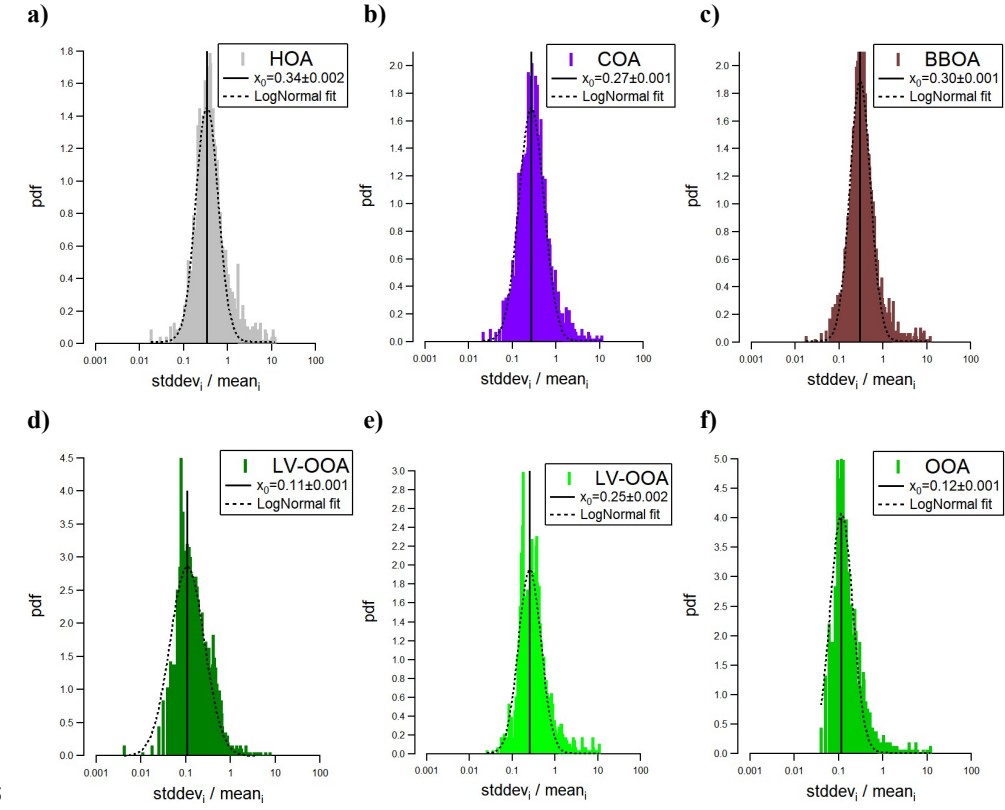


**Figure 8: Probability density functions for the *PMF_error* of the six factors as a logarithmic representation on the x-axis.**
