# Peer review of "A new method for long-term source apportionment with timedependent factor profiles and uncertainty assessment using SoFi Pro: application to one year of organic aerosol data"

_Atmospheric Measurement Techniques, 2020_

## Referee Comment (RC1) · Anonymous Referee #2 · 13 Aug 2020

General Comments: The manuscript by Canonaco et al. developed a new method for long-term source apportionment with time-dependent factor profiles, which is a necessary piece of work for long-term field campaigns data. The seasonal variations of OA factors in urban background station were investigated. Overall, the paper is well written. I recommend acceptance for publication on AMT after minor revisions.

Specific Comments:

[Figure]

1, line 125: Why has the authors re-averaged the data into half-an-hour resolution instead of using the original one? If the reason is the amount of data, then why not just averaging the data into two-hour(or three-hour) resolution? Please elaborate.

2, line 391: What is the difference between the mass spectra of COA in May 2011-September 2011 (likely due to local barbecuing events) and the general mass spectra of COA in this study? Has other studies discussed the characterization of mass spectra of different cooking styles? Please compare it.

3, LV-OOA was only identified before ∼1/11/2011 in Fig. 3, but why did the f44 in LV-OOA appear throughout the sampling time in Fig.2? In addition, there is no (c) in Fig. 2.

4, "Spring 11/Fall 11" in table2 should be "Spring 2011/Fall 2011".

Please also note the supplement to this comment:
https://amt.copernicus.org/preprints/amt-2020-204/amt-2020-204-RC1-supplement.pdf

―――――――――――――――――

---

## Referee Comment (RC2) · Anonymous Referee #1 · 11 Sep 2020

The study of Canonaco et al. reports a significant method development for improved PMF based source apportionment by aerosol mass spectrometers and crucially aimed at long time series of measurements badly lacking in literature. Recommendations and Conclusions are fairly discussed and well balanced which should help future researchers in properly using the method. The paper is very well written, easy to follow and should be accepted after addressing mostly minor comments.

[Figure]

Comments

Line 28. Past tense is more appropriate for the efforts in the past.

Line 39. ...and slightly higher mean concentrations...

Line 86. agricultural waste/residue burning.

Line 125. Average of the average (two-level averaging) reduces the weight of outliers and should generally be avoided, because it makes two-level averaged data not strictly compatible with one-level averaged tracers. Please elaborate on tracer data in relation to that. It is compounding of the fact that arithmetic averaging should not be applied to atmospheric variables in general (see later comment)

Line 144. biomass burning impact

Line 183. The resultant uncertainty of individual uncertainties can be calculated by the square root of squared sum, i.e. three individual uncertainties of 10%, result in 17%. So the resultant uncertainty will always be higher, not "might be slightly outside the defined a-range".

Line 202. Section 0 typo here and later several times. Then Line 305.

Line 212. missing dot

Line 326. Criterion of highest possible correlation coefficient and maximal data coverage are working against each other, so must be a compromise. What was it? It is not clear why 0.6 or 0.8 is best and what data coverage does it correspond to?

Line 333. If COA is well established it should peak every single day just like traffic factor during rush hour. If COA was not resolved, maybe its not very real. I was always concerned about this factor being a combination of true COA and being a waste basket for increasingly processed aerosol during midday when photochemical activity is at its highest. That is why tracer m/z as in BBOA case would yield much more credible approach.

Line 375. Given the fact that aerosol properties are lognormally distributed due to fundamental principles, using arithmetic averages is not appropriate. The study is very much grounded in mathematics and statistics where proper usage of terms is not only expected but mandatory. I understand that historically inappropriate usage is continuing forever. When noted the issue is ignored while when demanded is considered harsh.

Line 392. If COA spikes are barbecue related do they all occur during weekend as barbecues are rare during weekdays.

Line 429. ...as the problematic data yields eBCxb concentrations near zero anyway...

Line 461. ...likely indicating significant impact of biomass burning.

Line 492. ...last third of the study period...

Line 502. Here is an example of mixing together lognormal and normal (sigma and mean) distributions.

Line 535. ...to achieve complete apportionment.

Figure 2. b) typed twice instead of c)

Figure 3. . . .clogging of ACSM inlet orifice.

Figure 5. . . .truncated at m/z 125?

Figure 6. ...of important m/z tracers.
* * *

---

## Author Comment (AC1) · 3 Nov 2020

The study of Canonaco et al. reports a significant method development for improved PMF based source apportionment by aerosol mass spectrometers and crucially aimed at long time series of measurements badly lacking in literature. Recommendations and Conclusions are fairly discussed and well balanced which should help future researchers in properly using the method. The paper is very well written, easy to follow and should be accepted after addressing mostly minor comments.

*We thank the reviewer for this very positive feedback. We are also convinced that this study will be of help for future long-term source apportionment studies.*

Comments

Line 28. Past tense is more appropriate for the efforts in the past.

*"leads" replaced by "led"*

Line 39. ...and slightly higher mean concentrations...

*"and" has been added in front of "slightly higher mean concentrations…"*

Line 86. agricultural waste/residue burning.

*"waste/residue" has been added in front of "burning"*

Line 125. Average of the average (two-level averaging) reduces the weight of outliers and should generally be avoided, because it makes two-level averaged data not strictly compatible with one-level averaged tracers. Please elaborate on tracer data in relation to that. It is compounding of the fact that arithmetic averaging should not be applied to atmospheric variables in general (see later comment)

*The data was re-averaged to 30 minutes due to its rather noisy nature and we were therefore more concerned to extract a stable signal for the PMF analysis. The external tracers are compatible with the ACSM data, as they had a much higher time resolution (one-minute averages) and were then post-averaged to the ACSM time stamp. The sentence describing the average has been updated accordingly. "The data was re-averaged to 30 min to obtain higher signal to noise ratios for ME-2 analysis".*

Line 144. biomass burning impact

*"influence" has been replaced by "impact"*

Line 183. The resultant uncertainty of individual uncertainties can be calculated by the square root of squared sum, i.e. three individual uncertainties of 10%, result in 17%. So the resultant uncertainty will always be higher, not "might be slightly outside the defined a-range".

*What is described in the text is not the uncertainty but the allowed variation in % for a single m/z during the PMF iteration. The text never reports the expression "uncertainty", so we believe that there is no action we should do here.*

Line 202. Section 0 typo here and later several times. Then Line 305.
Line 212. missing dot

*All typos corrected.*

Line 326. Criterion of highest possible correlation coefficient and maximal data coverage are working against each other, so must be a compromise. What was it? It is not clear why 0.6 or 0.8 is best and what data coverage does it correspond to?

*The correlation coefficient ($R_{Pearson}$) of 0.8 resulted from a previous seasonal resampling analysis and at first we tried to apply this as a threshold. The problem was that this led to a large amount (10 % and more) of non-modeled time points ($n_{non-modeled}$). Hence, we performed a sensitivity analysis on $R_{Pearson}$ by systematically lowering it until the amount of $n_{non-modeled}$ was negligible. This was achieved for $R_{Pearson}$ = 0.6. The main text reports this already around line 326. So we don't think there is need for further action here.*

Line 333. If COA is well established it should peak every single day just like traffic factor during rush hour. If COA was not resolved, maybe its not very real. I was always concerned about this factor being a combination of true COA and being a waste basket for increasingly processed aerosol during midday when photochemical activity is at its highest. That is why tracer m/z as in BBOA case would yield much more credible approach.

*Evidence for cooking contributions does not only come from the regular presence of the lunch peak during the weekdays, as it was also seen in the previous seasonal analysis of this dataset (Canonaco et al., SoFi, an IGOR-based interface for the efficient use of the generalized multilinear engine (ME-2) for the source apportionment: ME-2 application to aerosol mass spectrometer data, Atmos. Meas. Tech., 6, 3649–3661, 2013, 2013 and Canonaco et al., Seasonal differences in oxygenated organic aerosol composition: implications for emissions sources and factor analysis, Atmos. Chem. Phys., 15, 6993–7002, 2015), but also from the fact that the cooking factor has the characteristic fingerprint of cooking, i.e., the series m/z 41, 43, 55, 57, 69, 71, etc. with a ratio above one for m/z 55 and 57, contrary to traffic (e.g., Liu et al., Primary and secondary organic aerosol from heated cooking oil emissions, Atmos. Chem. Phys., 18, 11363–11374, 2018). Moreover, this fingerprint has shown to be more pronounced with higher contributions close to restaurants (Elser et al., High contributions of vehicular emissions to ammonia in three European cities derived from mobile measurements, Atmospheric Environment, 175, 210-220, 2018).*

Line 375. Given the fact that aerosol properties are lognormally distributed due to fundamental principles, using arithmetic averages is not appropriate. The study is very much grounded in mathematics and statistics where proper usage of terms is not only expected but mandatory. I understand that historically inappropriate usage is continuing forever. When noted the issue is ignored while when demanded is considered harsh.

*The reviewer is right, aerosol properties are lognormally distributed. However, the source apportionment was conducted with no size-separation, i.e., non-refractory aerosol particles with an aerodynamic particle diameter with less than 1 micron (NR-PM1) were simply averaged together, based on the working principle of the ACSM. This truncates the relationship between the reported ACSM mass and the lognormal aerosol mass distribution. Moreover, when performing bootstrap and a value analysis for the PMF replicates, given their random resampling nature, the replicates will be normally distributed. Hence, using average and variance to describe the combination of accepted PMF runs is legitimate and we don't think it's necessary to take some action in this respect.*

Line 392. If COA spikes are barbecue related do they all occur during weekend as barbecues are rare during weekdays.

*These spikes occur more frequently during the weekends, but they also occur during the week. Barbecuing for dinner during warm days happens rather often in Switzerland.*

Line 429. ...as the problematic data yields eBCxb concentrations near zero anyway... Line 461. likely indicating significant impact of biomass burning.

*Both lines updated as recommended*

Line 492. ...last third of the study period...

*"campaign" has been replaced by "measurement".*

Line 502. Here is an example of mixing together lognormal and normal (sigma and mean) distributions.

*The reviewer mentions a possible mixing due to the expression of sigma and avg. in the error equation and its final lognormal fit. The distribution of the replicates per time point follows a normal distribution and hence the use of sigma and avg. is legitimate.*
*The PMF error reported in this study was based on the following distribution:*

$$PMF_{error} = \frac{100}{2 \cdot n} \cdot \sum_{i=0}^{n} \left(\frac{\sigma}{avg}\right)_i$$

*where $\sigma$ is the standard deviation and avg is the mean value of all replicates of a time-point $i$.*
*This resulting distribution, which is a constructed distribution using the statistics of the distribution of the replicates, turns out to be lognormally distributed and consequently a lognormal fit had to be used to best describe its shape. The types of distributions are not inadvertently mixed and therefore we don't see the necessity for further action in this respect.*

Line 535. to achieve complete apportionment.

*Updated as recommended*

Figure 2.  b) typed twice instead of c)

*Corrected*

Figure 3. ... clogging of ACSM inlet orifice.

*Updated as recommended*

Figure 5. ... truncated at m/z 125?

*No, the mass spectrum for the source apportionment was only up to 125.*

Figure 6. ...of important m/z tracers.

*Updated as recommended*

---

## Author Comment (AC2) · 3 Nov 2020

General Comments: The manuscript by Canonaco et al. developed a new method for long-term source apportionment with time-dependent factor profiles, which is a necessary piece of work for long-term field campaigns data. The seasonal variations of OA factors in urban background station were investigated. Overall, the paper is well written. I recommend acceptance for publication on AMT after minor revisions.

*We thank the reviewer for this very positive feedback. We are also convinced that this study will be of help for future long-term source apportionment studies.*

Specific Comments:

1, line 125: Why has the authors re-averaged the data into half-an-hour resolution instead of using the original one? If the reason is the amount of data, then why not just averaging the data into two-hour(or three-hour) resolution? Please elaborate.

*Averaging the data to 30 minutes represents a trade-off between a better signal to noise ratio and the presence of a sufficiently resolved diurnal cycle (here one-hour resolution), crucial for the source validation step.*

2, line 391: What is the difference between the mass spectra of COA in May 2011-September 2011 (likely due to local barbecuing events) and the general mass spectra of COA in this study? Has other studies discussed the characterization of mass spectra of different cooking styles? Please compare it.

*The ratio of m/z 55 to 57 as well as m/z 43 and 44 vary in the range of a few percentages, but there are no systematic or seasonal changes. Hence, for this study not much can be concluded for the temporal variability of the COA fingerprint. The largest seasonal change reported in this study is mainly for OOA, SV-OOA in particular.*

3, LV-OOA was only identified before 1/11/2011 in Fig. 3, but why did the f44 in LV-OOA appear throughout the sampling time in Fig.2? In addition, there is no (c) in Fig. 2.

*f44 in Figure 2 is for both, i.e., LV-OOA and OOA. Hence, during the warm seasons f44 in Figure 2 is for the LV-OOA factor, whereas in winter it is for OOA only. In Fig. 2 the y axis reads now: "f44 in LV-OOA/OOA".*

*c) has been corrected in Fig. 2*

4, "Spring 11/Fall 11" in table2 should be "Spring 2011/Fall 2011".

*Corrected*

Please also note the supplement to this comment:
https://amt.copernicus.org/preprints/amt-2020-204/amt-2020-204-RC1-supplement.pdf

*The supplement contained the exact same review as already reported here.*